# Femtosecond Optical Nonlinearity of Nanodiamond Suspensions

**Gennady M. Mikheev** [1,*] , **Viatcheslav V. Vanyukov** [2], **Tatyana N. Mogileva** [1], **Konstantin G. Mikheev** [1], **Alexander N. Aleksandrovich** [1], **Nicholas A. Nunn** [3,4] **and Olga A. Shenderova** [3]

1 Udmurt Federal Research Center, Institute of Mechanics, Ural Branch of the Russian Academy of Sciences, 426067 Izhevsk, Russia; mogileva@udman.ru (T.N.M.); k.mikheev@udman.ru (K.G.M.); alexan@udman.ru (A.N.A.)
2 Department of Physics and Mathematics, Institute of Photonics, University of Eastern Finland, 80101 Joensuu, Finland; viatcheslav.vanyukov@uef.fi
3 Adamas Nanotechnologies, Inc., Raleigh, NC 27617, USA; nnunn@adamasnano.com (N.A.N.); oshenderova@adamasnano.com (O.A.S.)
4 Department of Chemistry, North Carolina State University, Raleigh, NC 27695, USA
* Correspondence: mikheev@udman.ru; Tel.: +7-34-1221-8955

**Abstract:** High pressure-high temperature (HP-HT) nanodiamonds and detonation nanodiamonds have unique optical properties and are promising materials for various applications in photonics. In this work, for the first time, comparative studies of the nonlinear optical properties of aqueous suspensions of HP-HT and detonation nanodiamonds under femtosecond laser excitation are performed. Using the z-scan technique, it was found that for the same laser pulse parameters HP-HT nanodiamonds exhibited optical limiting due to two-photon absorption while detonation nanodiamonds exhibited saturable absorption accompanied by short-term optical bleaching, revealing the different electronic-gap structures of the two types of nanodiamonds. The saturable absorption properties of detonation nanodiamonds are characterized by determining the saturable and non-saturable absorption coefficients, the saturation intensity, and the ratio of saturable to non-saturable losses. The nonlinear absorption in HP-HT nanodiamonds is described with the nonlinear absorption coefficient that decreases with decreasing concentration of nanoparticles linearly. The results obtained show the possibility of using aqueous suspensions of nanodiamonds for saturable absorption and optical limiting applications.

**Keywords:** nanodiamonds; nonlinear optics; nonlinear refraction; nonlinear absorption; two-photon absorption; saturable absorption; optical limiting





## 1. Introduction

From the invention of lasers to the present time, the intensive study of the nonlinear optical properties of various materials has continued. The rapid development of nanotechnology in recent years has led to a wide variety of nanomaterials, including nanocarbons, which have nonlinear optical properties and that are of great interest for various applications. To date, there are a large number of works devoted to the study of the nonlinear optical properties of carbon nanotubes, graphene, and various composites based on them. These works are aimed at investigating, e.g., optical limiting (a nonlinear decrease of the transmitted light intensity) [1] and saturable absorption (a nonlinear increase of the transmitted light intensity) [2] to develop and create optical limiters of nanosecond laser pulses [3–13] and passive laser shutters for generating picosecond and femtosecond laser pulses [14–24], respectively. Nanodiamond, being another form of nanocarbon material, is also attractive due to its unique properties [25–33]. Among the various methods for producing nanodiamonds [27,34], the most prevalent are the detonation synthesis method [35] and the method of grinding diamonds synthesized at high pressure and high temperature (HP-HT) [36].

The nonlinear optical properties of detonation nanodiamonds and various complexes based on them have been extensively studied. Similar to carbon nanotubes and graphene, detonation nanodiamonds have been studied for the development and creation of nonlinear optical filters operating in a wide wavelength range to protect the eyes and sensitive optical components from the damaging effect of powerful laser pulses of nanosecond duration. It was found that in addition to nonlinear absorption, the nonlinear scattering of light has a significant effect on the nonlinear decrease in the optical transmittance with increasing incident light fluence in the nanosecond time domain [37–41]. Nonlinear absorption and nonlinear scattering have been found, depending on the nanodiamond particles size and concentration in suspensions [42–44]. It was also found that the functionalization of detonation nanodiamonds with other nanoparticles and complexes suitable for this purpose leads to a further decrease in the nonlinear transmittance coefficient [41,45–48]. It has been also shown that aqueous suspensions of detonation nanodiamonds exhibit a saturable absorption (a short-term increase in the transmittance) under nanosecond [49] and femtosecond [50] excitations. Despite the number of studies of nonlinear optical properties of detonation nanodiamonds, to the best of our knowledge, there is only one work on the nonlinear optical properties of HP-HT nanodiamonds. In that work, the observation of optical limiting in an aqueous suspension of HP-HT nanodiamonds under excitation by nanosecond laser pulses at a wavelength of 532 nm was reported [51]. This is of interest for further investigation of the nonlinear optical properties of HP-HT nanodiamonds by exploring the femtosecond time domain and to compare the nonlinear optical response of those with the detonation nanodiamonds under the same experimental conditions.

In this work, we show that aqueous suspensions of HP-HT nanodiamonds exhibit nonlinear optical properties in the femtosecond time domain. Moreover, we show that aqueous suspensions of detonation and HP-HT nanodiamonds with an average size of ~10 nm possess different nonlinear optical properties under femtosecond laser excitation. Specifically, suspensions of HP-HT nanodiamonds exhibit optical limiting originating from the two-photon absorption, while suspensions of detonation nanodiamonds demonstrate optical self-bleaching due to saturable absorption.

## 2. Materials and Methods

HP-HT nanodiamonds were obtained from Van Moppes, Geneva, Switzerland. The HP-HT nanodiamond particles are monocrystals with a substitutional nitrogen content of ~100 ppm. The purification of nanodiamonds from carbon was carried out by processing in the air at a temperature of 500 °C for two hours, followed by treatment in hydrochloric acid at a temperature of 70 °C for two hours. The nanoparticles were dispersed in deionized water at 1.65 wt% concentration and the suspensions were stored in plastic vessels.

Detonation nanodiamonds were synthesized by an explosion of an oxygen-deficient explosive mixture of trinitrotoluene and hexogen at a weight ratio of 1:1 in a closed steel chamber. The resulting detonation soot contains the nanodiamond particles (more than 30%), the other allotropic states of carbon, and various metallic impurities. Purification of the diamond soot in the mixture at the high temperature reduces the amount of impurities down to 1 wt%. These detonation nanodiamonds received from a vendor were additionally purified at Adamas Nanotechnologies with HCl, reducing the metal content to 0.4 wt%. Detonation synthesis leads to the formation of nanodiamonds with a primary particle size of about 5 nm. The natural agglomeration of nanodiamonds leads to the formation of solid particles of much larger sizes. To obtain nanoparticles with a given average size the purified nanodiamonds were suspended in deionized water by sonication and processed in a planetary mill for 4 h using zirconia beads. Additional $sp^2$-carbon from milling was removed by treatment at 400 °C in the air for 3 h. After this high-temperature treatment, the resulting product was resuspended in deionized water at 1% *w/v* by sonication. Centrifugation at 25,000× *g* forces was used to extract nanodiamond particles with an average diameter of ~10 nm. The nitrogen content in the detonation nanodiamonds is ~10,000 ppm.

The major differences in HP-HT and detonation nanodiamonds are in the size of the primary particles, the presence of aggregates consisting of few primary particles in detonation nanodiamonds and nitrogen content. While treatments to reduce $sp^2$-carbon content were performed, residual $sp^2$-carbon is still present in both types of nanodiamond particles, with a higher content expected in detonation nanodiamonds due to the smaller size of primary particles.

For surface and particle size analysis, small droplets of fabricated suspensions of HP-HT and detonation nanodiamonds were placed on a flat glass surface and dried to form a uniform film. The topography images of the nanodiamond films were acquired with an NTEGRA Probe Nanolaboratory atomic-force microscope (AFM) in semicontact mode. The particle size distribution was attained by processing the atomic-force microscope images with Image Analysis 3.5.0 software (NT-MDT Spectrum Instruments, Zelenograd, Russia) based on the combination of the correlation analysis and the method of sections at a certain relative height. The AFM images and the particle size distribution of the nanodiamond films are shown in Figure 1. The performed measurements reveal that more than 75% of the HP-HT and detonation nanodiamonds have sizes in the ranges of 9–20 and 6–13 nm, respectively.

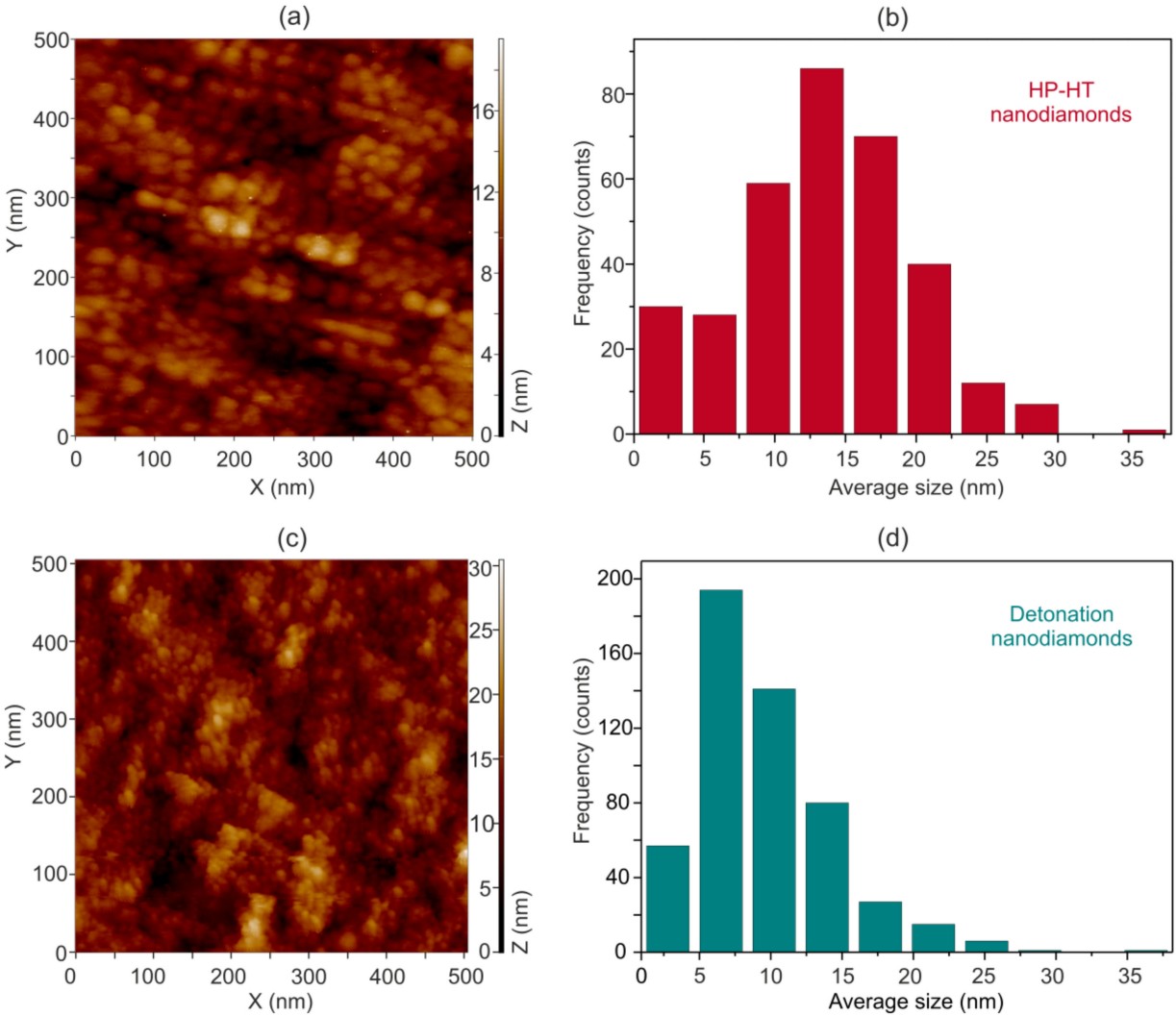

**Figure 1.** Atomic force microscopy images of (**a**) HP-HT and (**c**) detonation nanodiamond films. The image processing reveals the HP-HT and detonation nanodiamond particle size distributions shown in figures (**b**) and (**d**), correspondingly.

Raman spectra of the attained nanodiamond films were recorded using Horiba LabRam HR800 Raman spectrometer (Lille, France) with a 632.8 nm laser. A 100× objective with the attenuation of the incident radiation (neutral filter D1) was used to provide an intensity (less than about 5 kW/cm$^2$) below the graphitization of nanodiamond material at 632.8 nm [29,52]. The measured Raman spectra of detonation and HP-HT nanodiamond films show pronounced peaks with frequency shifts of 1329 and 1330 cm$^{-1}$, respectively (see Figure 2a). Taking into account the known dependences of the Raman shifts of carbon nanomaterials on the excitation wavelength [53] and the nanoparticle size [54], one can attribute these peaks to nanodiamonds (sp$^3$-form of carbon). The decomposition of the spectra presented in Figure 2a shows that the full width at half maximum (FWHM) of diamond lines of HP-HT and detonation nanodiamonds are 8 and 22 cm$^{-1}$, respectively. It is noteworthy that along with the specified peaks, the Raman spectra exhibit broad luminescence bands caused by defects of various natures contained in HP-HT and detonation nanodiamonds [29,55].

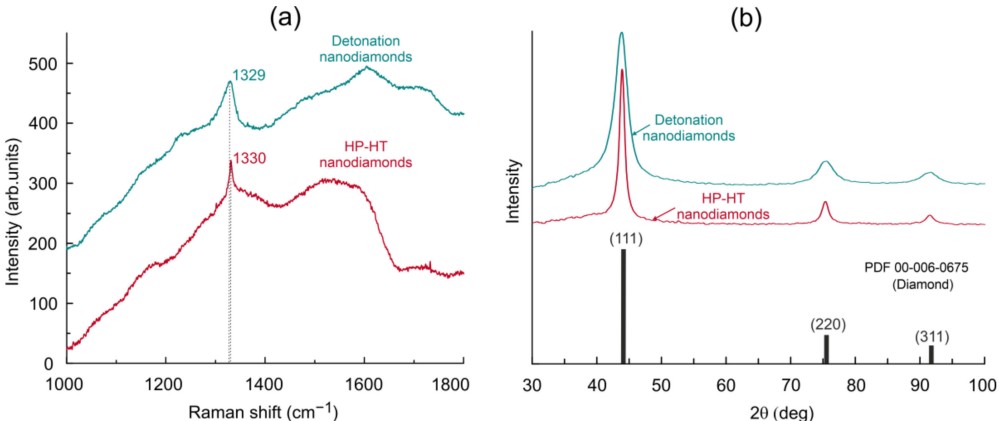

**Figure 2.** (**a**) Raman spectra of HP-HT and detonation nanodiamonds film and (**b**) X-ray diffraction patterns of HP-HT and detonation nanodiamond powders. The X-ray diffraction pattern of diamond (Powder Diffraction File 00-006-0675) is presented for a reference.

For the X-ray diffraction studies, nanodiamond powders were obtained by drying the suspensions at room temperature and subsequent mechanical grinding. When obtaining diffraction patterns, the studied powders were placed on a glass substrate. The diffraction pattern of the substrate without powder was recorded separately. To obtain the diffraction pattern of the studied powder, the corresponding subtraction of the substrate response from the experimental data was performed.

The crystalline structure of the nanodiamonds was studied using Bruker D2 PHASER X-ray diffractometer with a copper-based X-ray tube generating radiation at a wavelength of 0.1541 nm (K$_{\alpha 1}$ line). The diffraction patterns of the detonation and HP-HT nanodiamonds powders (see Figure 2b) contain three solitary peaks in the range of 2θ variation from 25° to 100°. These peaks are observed at 2θ angles of 43.9°, 75.3°, and 91.5°. They correspond to X-ray diffraction on the (111), (220), (311) planes of diamond crystallites with interplanar spacings of d$_{111}$ = 0.2060 nm, d$_{220}$ = 0.1261 nm, and d$_{311}$ = 0.1075 nm, respectively, where the subscripts at d denote crystallographic planes. Note that according to the reference data of diamond (see Figure 2b), the intensities of the diffraction lines of monochromatic radiation on the (111), (220), (311) planes are equal to 100, 25, and 16 arb. units, respectively (see Powder Diffraction File (PDF) 00-006-0675). Figure 2b shows that for both diffraction patterns, the ratio between peak amplitudes at angles 2θ equal to 43.9°, 75.3°, and 91.5° is the same. Processing the measured X-ray diffraction pattern using TOPAS 4.2 software revealed that the mean crystallite sizes (LVol-IB) of the HP-HT and detonation nanodiamond particles are 8.6 and 2.7 nm, respectively. Thus, the

measurements on an X-ray diffractometer show that all the samples under study consist of diamond material.

For the optical measurements, suspensions of both types of nanodiamonds were prepared at three concentrations. Figure 3 shows the optical densities of aqueous suspensions of HP-HT and detonation nanodiamonds at three different concentrations measured in a 1 mm thick optical quartz cell. The measurements were performed using a PerkinElmer Lambda 650 (Shelton, WA, USA) two-beam spectrophotometer relative to the same 1 mm-thick quartz cell filled with distilled water. It can be seen that for the suspensions of both types of nanodiamonds, the optical density decreases monotonically with an increase of light wavelength which is the characteristic of nanodiamonds [29,32,49,56,57]. For the suspensions of HP-HT nanodiamonds, the increase in optical density with decreasing wavelength is more pronounced than for the suspensions of detonation nanodiamonds. This can be explained by stronger light scattering in the suspension of HP-HT nanodiamonds caused by the following reasons: (i) the slightly larger average nanoparticle size of the HP-HT nanodiamond suspension compared to the average nanoparticle size of the detonation nanodiamond suspension; (ii) the larger "effective" refractive index of HP-HT nanodiamonds compared to the "effective" refractive index of detonation nanodiamonds due to the difference in their crystal structures. For both types of nanodiamonds the optical density measured at different wavelengths is proportional to the concentration (see insets in Figure 3a,b).

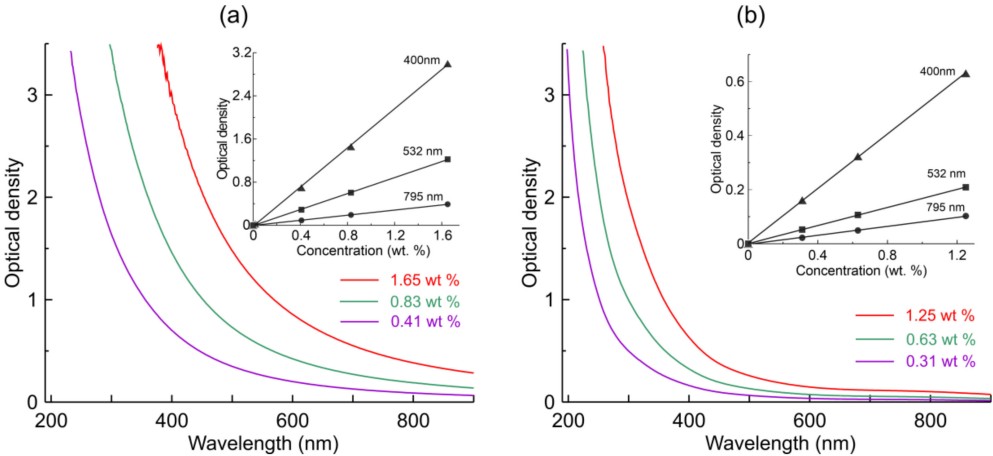

**Figure 3.** The optical density of the (**a**) HP-HT and (**b**) detonation nanodiamond suspensions in 1 mm thick quartz cell. Insets show the corresponding optical density as a function of concentration measured at three different wavelengths of 400, 532, and 795 nm.

Studies of the nonlinear optical properties of nanodiamond suspensions were carried out using the z-scan technique [58] with femtosecond linearly polarized laser pump (wavelength 795 nm, pulse duration 120 fs, and the repetition rate of 1 kHz). In the experiments, simultaneous measurements of transmittance in the closed- and open-aperture configurations were performed (Figure 4). The laser radiation was focused on a cell with a focal length of 75 mm. The cell had 1 mm-thick quartz walls with a 1 mm opening distance. The laser beam waist radius was $w_0 = 12.9$ μm, which corresponds to the Rayleigh length of $z_0 = 0.65$ mm ($z_0 = \pi w_0^2/\lambda$, where $\lambda$ is the wavelength). The scanning of the cell was performed along the optical z-axis of the focused laser beam using computer numerical control with a step of 0.1 mm with multiple averaging of the pulse energies $E_{oa}$ and $E_{ca}$ recorded using "open-aperture" and "closed-aperture" photodetectors, respectively (see Figure 4).

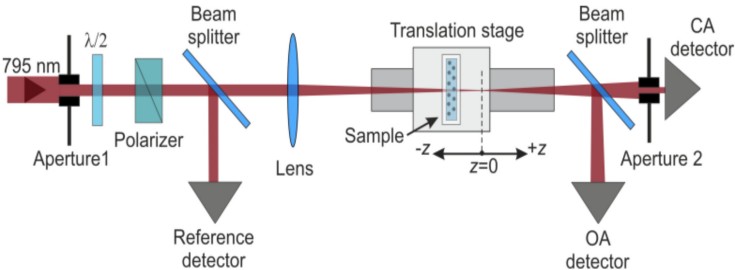

**Figure 4.** Sketch of the z-scan experimental setup.

In the experiments, the dependences of the nonlinear transmittance coefficients $T_{oa}(z)$ = $E_{oa}(z)/E_{in}$ (with an open-aperture) and $T_{ca}(z) = E_{ca}(z)/E_{in}$ (with a closed-aperture) were measured during scanning of the cell along the z-axis, where $E_{in}$ is the incident laser pulse energy measured using a reference photodetector. This made it possible to determine the dependences of the normalized nonlinear transmittance coefficients $T_{oa,n}(z) = T_{oa}(z)/T_{oa}(z = \infty)$ and $T_{ca,n}(z) = T_{ca}(z)/T_{ca}(z = \infty)$ with open- and closed-apertures respectively, where $T_{oa}(z = \infty)$ and $T_{ca}(z = \infty)$ are the optical transmittance coefficients at open- and closed-apertures, respectively, obtained far from the beam waist, i.e., far from z = 0. The energy of laser pulses incident on the cell was varied using a polarizer and a half-wave plate. To visualize nonlinear refraction in the experiments, the ratio of nonlinear transmittance obtained in a closed- aperture to the data obtained in an open-aperture $t_{nr}(z) = T_{ca,n}(z)/T_{oa,n}(z)$ was determined [59].

## 3. Results

The dependences $T_{oa,n}(z)$ and $t_{nr}(z)$ obtained for a cell filled with distilled water (a,b) and an empty cell (c,d) at $E_{in}$ = 200 nJ are presented in Figure 5a–d. The dependence $T_{oa,n}(z)$ obtained for a cell with distilled water shows a small dip in the vicinity of $z/z_0$ = 0 indicating a weak nonlinear absorption. At the same time, the experimental data $T_{oa,n}(z)$ obtained for an empty cell (Figure 5c) shows no corresponding dip. This means that the observed weak nonlinear absorption for a cell with distilled water occurs because of distilled water and not because of the cell. The nonlinear absorption in distilled water originates from multi-photon absorption [60]. Comparison of the closed/open-aperture data for a cell filled with water (Figure 5b) and an empty cell (Figure 5d) shows that the observed nonlinear refraction leading to self-focusing of the femtosecond laser beam occurs mainly in the quartz walls of the optical cell. The observed cluster of dots at the vicinity of $z/z_0$ = 0 in Figure 5d arises from the quartz walls-air-quartz walls boundary and not affecting on the amplitude of the normalized transmittance. Closed/open-aperture measurements were also performed for both the studied types of nanodiamonds with the results shown in Figure 5e,f). Adding the nanoparticles to water has no noticeable impact on the nonlinear refraction of the suspensions as one can observe by comparing Figure 5b,d–f) indicating weak nonlinear refraction in both suspensions for given concentrations.

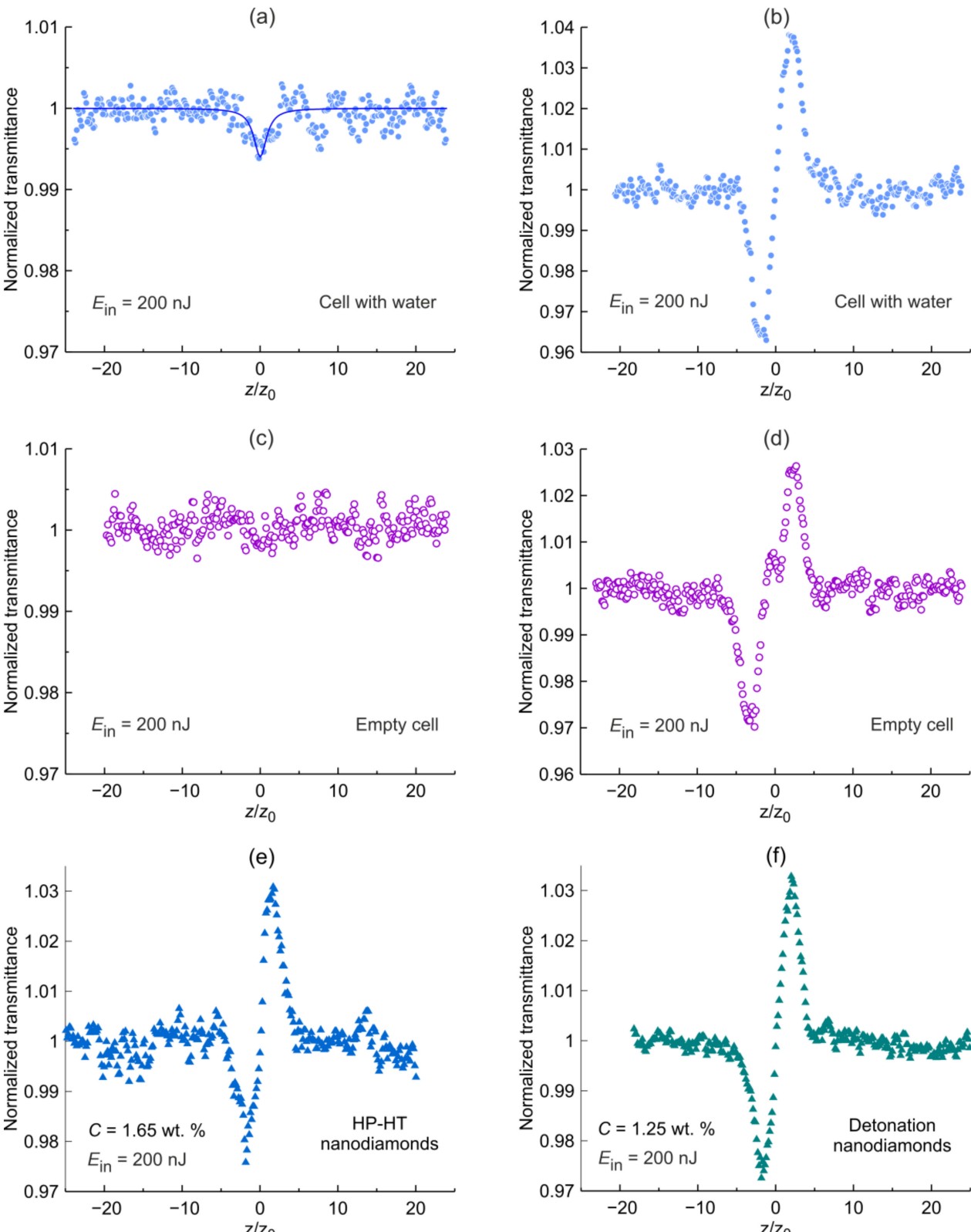

**Figure 5.** The normalized transmittance of the quartz cell with (**a**) distilled water and (**c**) an empty cell obtained in the open-aperture z-scan. The ratio of the nonlinear transmittance obtained in the closed to the data obtained in the open-aperture z-scan for the quartz cell (**b**) with distilled water, (**e**) with HP-HT nanodiamond suspension, (**f**) with detonation nanodiamond suspension, and (**d**) an empty cell. The measurements were performed at the incident laser pulse energy of 200 nJ.

The results of the open-aperture normalized transmittance measurements for aqueous suspensions of HP-HT nanodiamonds are shown in Figure 6. The measurements were performed at the same incident pulse energy of $E_{in}$ = 200 nJ. In order to reveal the dependence of the nonlinear optical properties, i.e., nonlinear absorption on the nanoparticle concentration C, we performed the measurements for the C ranging from 0.1 to 1 wt%. It is shown that the higher the concentration, the stronger the nonlinear absorption, the lower the transmittance at $z/z_0$. Even at a low concentration of 0.08 wt%, the nonlinear transmittance exhibits a visible dip. It is noteworthy that the dependencies shown in Figure 6 are symmetric relative to $z/z_0$ = 0 indicating prominent optical limiting properties of femtosecond laser pulses.

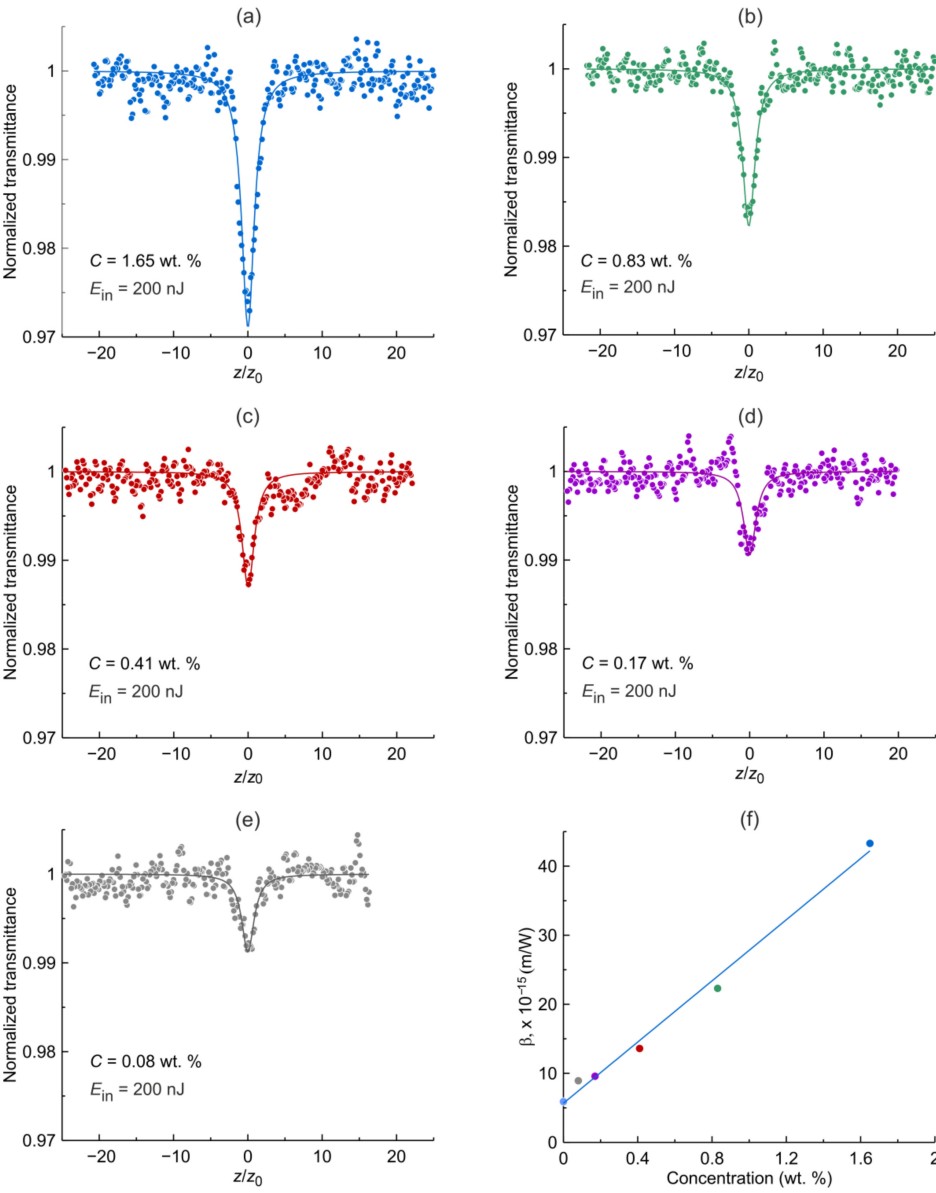

**Figure 6.** (**a–e**) Normalized transmittance of suspensions of HP-HT nanodiamonds with different concentrations recorded in the open-aperture z-scan. Dots correspond to the experimental data while solid lines represent the result of the fitting with Equation (2). Figure (**f**) represents the dependence of the calculated two-photon absorption coefficient β as a function of concentration C.

One can assume that the mechanism of optical limiting in suspensions of HP-HT nanodiamonds is two-photon absorption, in which the absorption coefficient $\alpha_{TPA}$ of the medium depends on the radiation intensity I according to the equation [61]

$$\alpha_{TPA}(I) = \alpha + \beta I, \tag{1}$$

where $\alpha$ and $\beta$ are the coefficients of linear and nonlinear absorption, respectively. This can be verified by approximating the experimental data presented in Figure 6 with the well-known equation (see [61]) derived to describe two-photon absorption for the open-aperture z-scan for a Gaussian laser beam and temporally Gaussian pulse:

$$T_{oa,n} = 1 + \sum_{m=1}^{\infty} \left[-q(z, t = 0)\right]^m / (m+1)^{3/2}, \tag{2}$$

where $q_0(z,\, t = 0) = \beta I_0(t = 0)L_{eff}/(1 + z^2/z_0^2)$, $L_{eff} = (1 - e^{-\alpha L})/\alpha$ , $\alpha = -lnT_0/L$, $I_0(t = 0) = E_{in}/[\pi w_0^2 \Delta t \frac{\sqrt{\pi}}{2\sqrt{ln2}}]$ (see [58]), $I_0(t)$ is the on-axis irradiance at focus (i.e., z = 0), $\Delta t$ is the pulse duration (FWHM) of a Gaussian laser pulse, $T_0$ is the linear transmittance of the studied sample with thickness L. Equation (2) with the first four terms from an infinite series of elements is in good agreement with the experimental z-scan data shown in Figure 6. The values of $\beta$ found as a result of the approximation of the experimental data, depending on the concentration of nanoparticles C, are shown in Figure 6f and can be expressed as $\beta(m/W) = (5.7 + 22.1 \times C(wt\%)) \times 10^{-15}$. It follows from the approximation, that the defined $\beta$ does not vanish when the concentration of nanoparticles in suspension is tending to zero confirming the earlier obtained z-scan data for a cell filled with distilled water (see Figure 5a). Based on the above, one can conclude that the two-photon absorption well describes the mechanism of the nonlinear behavior in aqueous suspensions of HP-HT nanodiamonds.

The results of the normalized transmittance recorded in the open-aperture z-scan for a fixed concentration of aqueous suspension of detonation nanodiamonds are shown in Figure 7. We have not presented the measurements for the various concentrations as for HP-HT nanodiamonds because those have been already reported [49]. Instead, measurements of the normalized transmittance with variation of the incident laser pulse energy have been performed. In contrast to HP-HT nanodiamonds, suspensions of detonation nanodiamonds exhibit an increase in the nonlinear transmittance with increasing incident light intensity. In the range of incident laser pulse energies, the experimental curves of $T_{oa,n}(z)$ presented in Figure 7 are symmetrical relative to the focal point $z/z_0 = 0$. This suggests that the observed phenomenon is not associated with laser-induced irreversible bleaching [62,63], but originates from the saturable absorption which has been extensively studied in graphene, carbon nanotubes [15,23,24,64,65] and has also been reported in detonation nanodiamonds with larger particle sizes under femtosecond excitation [50]. Similar to graphene and carbon nanotubes the observed saturable absorption in detonation nanodiamonds is characterized by a short-term decrease of its absorption coefficient during the passing of a light pulse of high intensity and caused by the electronic transition between two energy levels.

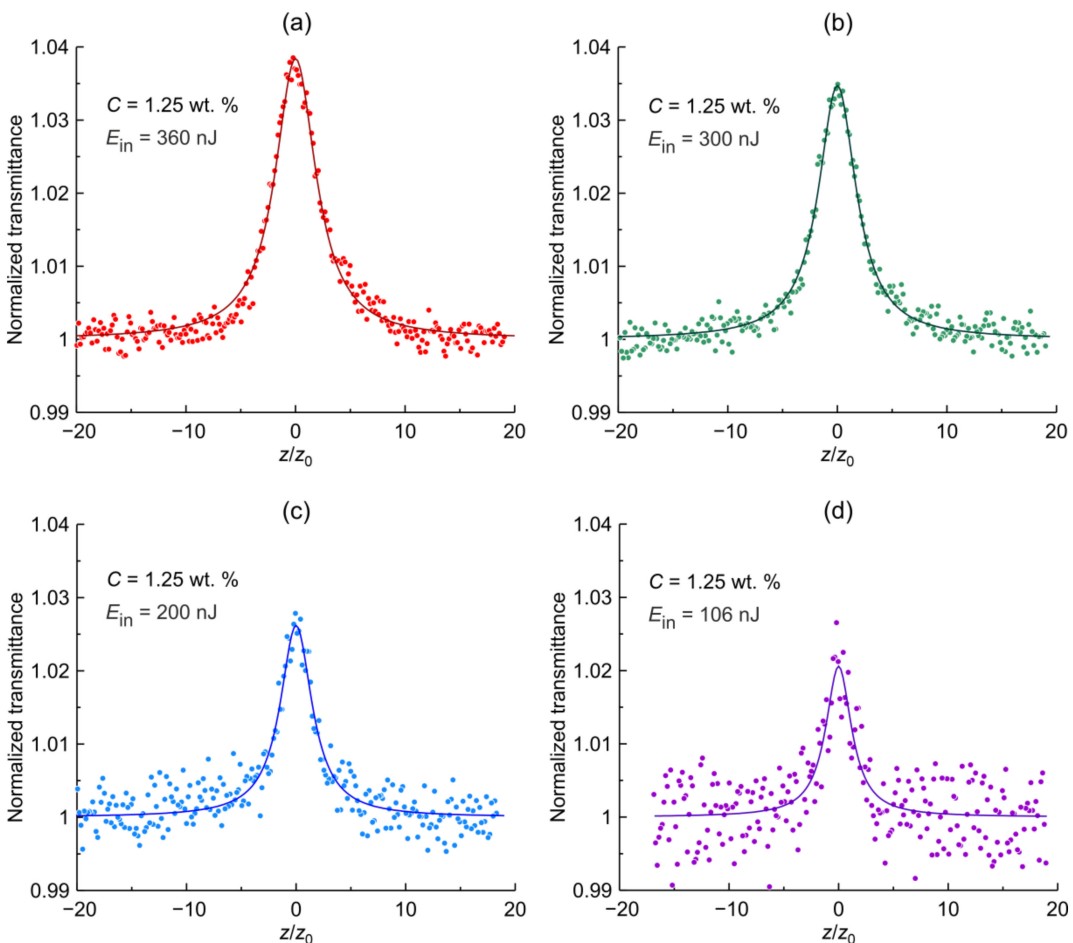

**Figure 7.** (**a–d**) Normalized transmittance of suspensions of detonation nanodiamonds obtained in the open-aperture z-scan for a fixed concentration of 1.25 wt% for various incident laser pulse energies ranging from 106 to 360 nJ. Dots represent the experimental data while solid lines represent the result of the fitting with Equation (5).

In accordance with [65,66], the absorption coefficient at saturable absorption is given by:

$$\alpha_{SA}(I) = \alpha_{ns} + \alpha_0/(1 + I/I_{sat}) \tag{3}$$

where $\alpha_{ns}$ is the linear absorption coefficient due to the presence of light losses in the medium that are not associated with saturable absorption, $\alpha_0$ is the coefficient characterizing the absorption in a two-level energy system, where the population moves under the action of resonant laser pumping, leading to equalization of the level populations at high intensities (saturation of absorption) [67], $I_{sat}$ is the saturation intensity, i.e., the intensity at which $\alpha_0$ is halved. It should be noted that the linear absorption coefficient $\alpha_{SA}(I = 0) = \alpha_{ns} + \alpha_0$, and at $I \gg I_{sat}$ $\alpha_{SA}(I \to \infty) = \alpha_{ns}$, is true, i.e., even at very high pump intensities, complete bleaching of the medium does not occur. The coefficients $\alpha_0$ and $\alpha_{ns}$ are commonly referred to as saturable and nonsaturable absorptions, respectively.

As was established above, the observed multiphoton absorption in pure distilled water is weak and therefore one can neglect its effect on the saturable absorption. In order to find the dependence of nonlinear transmittance on the incident intensity of a suspension of nanoparticles exhibiting saturable absorption the following differential equation should be solved:

$$dI/dz\prime = -\alpha_{SA}I, \tag{4}$$

which is describing the propagation of the light pulse in a nonlinear medium [61] with the absorption coefficient presented in the form (3). For this case, the exact solution of Equation (4) yields to [49]:

$$T_{oa,n} = \left[ \frac{I_{out}/I_{sat} + 1 + \alpha_0/\alpha_{ns}}{I_{in}/I_{sat} + 1 + \alpha_0/\alpha_{ns}} \right]^{-\alpha_0/\alpha_{ns}}, \tag{5}$$

where $I_{out}$ and $I_{in}$ are the output and incident radiation intensities. Saturable absorption leads to a slight increase in the intensity at the output of the cell (see Figure 7), therefore, the following expression is valid $I_{out} = T_0 \times I_{in}$. For a Gaussian beam during z-scan, the relation $I_{in} = I_0/(1 + (z/z_0)^2)$ is satisfied. Taking all this into account, one can arrive at the following equation describing the dependence of the normalized transmittance on the sample position:

$$T_{oa,n} = \left\{ \frac{T_0 \times A + (1 + B) \times \left(1 + (z/z_0)^2\right)}{A + (1 + B) \times \left(1 + (z/z_0)^2\right)} \right\}^{-B}, \tag{6}$$

where $A = I_0/I_{sat}$ and $B = \alpha_0/\alpha_{ns}$.

Equation (6) makes it possible to approximate the experimental dependences shown in Figure 7 with two parameters $A$ and $B$. Considering that $lnT_0 = -(\alpha_{ns} + \alpha_0) \times L$, where $T_0$ is the linear transmittance of the suspension, we arrive at the $I_{sat} = 110 \, \text{GW/cm}^2$, $\alpha_{ns} = 2.0 \, \text{cm}^{-1}$ and $\alpha_0 = 0.36 \, \text{cm}^{-1}$. It is seen from Figure 7 that Equation (6) with the defined parameters approximates the experimental data well. It should be noted that the obtained values of $\alpha_{ns}$ and $\alpha_0$ are defined for the nanoparticle concentration of C = 1.25 wt% and depend linearly on concentration [49], i.e., $\alpha(\text{cm}^{-1}) = 1.6 \times C \, (\text{wt}\%)$, $\alpha_0 \, (\text{cm}^{-1}) = 0.24 \times C \, (\text{wt}\%)$. This means that for both the HP-HT and detonation nanodiamonds, the nonlinear absorption coefficients, i.e., two-photon absorption β and saturable absorption $\alpha_0$ coefficients, linearly depend on the nanoparticles concentration. The obtained $\alpha_{ns}$ and $\alpha_0$ are similar to those obtained for the single-digit detonation nanodiamonds [49]. The attained saturation intensity $I_{sat}$ which is concentration independent [49] is the same order as for the aqueous suspensions of detonation nanodiamonds with an average particle size of 50 nm under the femtosecond pulse excitation [50] but $11 \times 10^3$ times higher than that for the single-digit detonation nanodiamonds under the nanosecond pulse excitation [49]. A similar difference in saturation intensities in nanosecond and femtosecond time domains has been observed in a graphene polymer composite [68].

Another important parameter characterizing the saturable absorption properties and the applicability of the materials for the passive mode-locking is the ratio of the saturable to non-saturable losses [65]:

$$R = \{\exp(-\alpha_{ns}L) - \exp[-(\alpha_{ns} + \alpha_0)L]\} / [1 - \exp(-\alpha_{ns}L)], \tag{7}$$

which should be as high as possible. Since the products $\alpha_{ns}L$ and $\alpha_0L$ are small, we arrive at $R = \alpha_0/\alpha_{ns} = 0.18$ which is lower than for graphene and single-walled carbon nanotubes ($R \approx 1$) used for passive mode-locking and Q-switching [65].

It should be added that in our experiments, despite the nonlinear absorption, the nonlinear refraction in the nanodiamond suspension due to heat absorption does not manifest itself. This is explained by the fact that the nonlinear absorption observed in the experiments is weak. For example, in the HP-HT nanodiamond suspension at a nanoparticle concentration of C = 1.65 wt% at a laser pulse energy of 200 nJ, the nonlinear absorption is only about 3% (see Figure 6a), and in the suspension of detonation nanodiamonds at the concentration of nanoparticles C = 1.25 wt% at the energy of laser pulses 360 nJ the nonlinear absorption does not exceed 4%.

## 4. Discussion

Comparing the obtained results of the nonlinear optical properties of HP-HT and detonation nanodiamond suspensions in the femtosecond time domain one can conclude that they have similar nonlinear refraction but different nonlinear absorption properties. The nonlinear refraction is weak for both types of nanodiamonds and has no impact on the nonlinear refraction of the aqueous suspension with a nanoparticle concentration of about 1 wt%. Under the same experimental conditions, two-photon absorption occurs in suspensions of HP-HT nanodiamonds leading to a decrease in the transmittance, while saturable absorption appears in an aqueous suspension of detonation nanodiamonds leading to self-bleaching indicating that the electronic structures of HP-HT and detonation nanodiamonds differ from each other significantly. It is well known that the energy of the bandgap $E_g$ of a pure bulk diamond is 5.4 eV and as a result there is no optical absorption in such material. The photon energy of the femtosecond laser we used is 1.56 eV, which is about 3.5 times less than $E_g$. This means that one-photon or two-photon absorptions, as well as saturable absorption at a wavelength of 795 nm in pure bulk diamond, are impossible. However, as can be seen from Figure 3, aqueous suspensions of HP-HT and detonation nanodiamonds are semitransparent in the optical range and their optical densities increase monotonically with decreasing wavelength. In accordance with [56], this is caused by light scattering (the cross-section of which decreases with an increase in the wavelength of incident radiation) and optical absorption that occurs in detonation nanodiamonds in the energy range of 1–2 eV. In a recent paper [69], it was shown that the characteristic extinction spectrum (see Figure 3) of detonation nanodiamonds with a size less than 100 nm is mainly determined by absorption rather than a scattering of light. The saturable absorption we observed in a suspension of detonation nanodiamonds at a photon energy of 1.56 eV is in agreement with these studies. The absence of saturable absorption in a suspension of HP-HT nanodiamonds indicates that there are no energy transitions that effectively absorb radiation at a wavelength of 795 nm. On the other hand, the observation of nonlinear absorption described by two-photon absorption shows that the suspensions of HP-HT nanodiamonds have an absorption band with transition energy of about 3.1 eV, which agrees with the optical density spectrum shown in Figure 3. The significant difference in the femtosecond nonlinear behavior of two types of nanodiamonds may be caused by their different structure.

Indeed, as established above, the average crystallite sizes of HP-HT and detonation nanodiamonds are 8.6 and 2.7 nm, respectively. Taking into account the sizes of the nanoparticles studied (see Figure 1), this means that HP-HT nanodiamonds are predominantly composed of one or two single crystallites, while detonation nanodiamond particles are conglomerates predominantly composed of two to five single crystallites. The difference in the structures of nanoparticles of the two types of nanodiamonds is also indicated by the difference in their Raman spectra presented in Figure 2a. The diamond Raman lines of the HP-HT and detonation nanodiamond spectra with frequency shifts of 1330 and 1329 cm$^{-1}$, respectively, differ from each other in line width and shape. The FWHM of the diamond line of detonation nanodiamonds is approximately 2.8 times greater than the corresponding value found for HP-HT nanodiamonds. In addition, the diamond line of detonation nanodiamonds is asymmetric with a more pronounced broadening towards lower wavenumbers. All these findings are in agreement with the literature data, and, according to Refs. [34] and [70], indicate a greater number of defects present in detonation nanodiamonds. Overall, monocrystalline HP-HT nanodiamonds have a uniform structure with a concentration of lattice defects (such as dislocations or twinning) lower than in detonation nanodiamonds containing a diamond core and various lattice defects, transient $sp^3/sp^2$ layer, and $sp^2$ surface shell that may carry various surface functional groups [70]. The $sp^2$-carbon content in detonation nanodiamonds is expected to be higher than in HP-HT nanodiamond due to the smaller size of primary particles. It should be noted that along with the presence of the $sp^2$-carbon [71], the appearance of energy transitions lower than $E_g$ in nanodiamond can be caused by the presence of diamond-like shells [72], the

dimer chains on the surface of diamond nanoparticles [56], as well as the numerous defects, vacancies, and impurities (see, for example, [29,73–75]). Independently of the origin of the two-photon absorption observed in an aqueous suspension of HP-HT nanodiamonds, it can be used for the optical limiting to protect sensitive optical components and eyes from the damaging effect of high-power femtosecond laser radiation with a wavelength of about 800 nm.

## 5. Conclusions

The nonlinear absorption of suspensions of HP-HT and detonation nanodiamonds at a wavelength of 795 nm of femtosecond excitation differ from each other significantly. At the same parameters of femtosecond laser pumping and experimental conditions, saturable absorption occurred in an aqueous suspension of detonation nanodiamonds accompanied by short-term nonlinear bleaching, while in an aqueous suspension of HP-HT nanodiamonds, two-photon absorption appeared, leading to optical limiting. The nonlinear absorption coefficient of the aqueous suspension of HP-HT nanodiamonds decreased linearly with the decreasing nanoparticle concentration. The important parameters characterizing the saturable absorption of detonation nanodiamonds in an aqueous suspension were determined. The results obtained show that the energy structures of HP-HT and detonation nanodiamonds differ from each other significantly. Aqueous suspensions of HP-HT nanodiamonds can be used for the optical limiting of high-power femtosecond laser pulses while suspensions of detonation nanodiamonds possess properties of the saturable absorber and can be employed for shortening the duration and to remove the pedestal ("tail") of the laser pulses.

**Author Contributions:** Conceptualization, G.M.M. and V.V.V.; methodology, G.M.M., V.V.V. and K.G.M.; software, V.V.V.; validation, G.M.M. and O.A.S.; formal analysis, V.V.V.; investigation, G.M.M., V.V.V., K.G.M., T.N.M. and A.N.A.; resources, N.A.N. and O.A.S.; data curation, T.N.M.; writing—original draft preparation, G.M.M. and V.V.V.; writing—review and editing, G.M.M., V.V.V., N.A.N. and O.A.S.; visualization, T.N.M.; supervision, G.M.M. and O.A.S.; project administration, G.M.M.; funding acquisition, G.M.M. All authors have read and agreed to the published version of the manuscript.

**Funding:** This research was funded by the Ministry of Education and Science of the Russian Federation, state registration number AAAA-A19-119021890083-0, by the Academy of Finland grants no. 298298 and 323053, Flagship Programme "Photonics Research and Innovation (PREIN)" decision no. 320165.

**Institutional Review Board Statement:** Not applicable.

**Informed Consent Statement:** Not applicable.

**Data Availability Statement:** The data presented in this study are available on request from the corresponding author.

**Acknowledgments:** The authors wish to thank S.G. Bystrov for his help in experiments. This study was performed using equipment of the Shared Use Center "Center of Physical and Physicochemical Methods of Analysis and Study of the Properties and Surface Characteristics of Nanostructures, Materials, and Products" UdmFRC UB RAS.

**Conflicts of Interest:** The authors declare no conflict of interest.

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
