# Peer review of "Femtosecond Optical Nonlinearity of Nanodiamond Suspensions"

_applsci, doi:10.3390/app11125455_

Round 1

Reviewer 1 Report

The authors report a study of non-linear refraction index and absorption in nanodiamond suspensions by means of open and closed aperture z-scan techniques. They compare HP-HT and detonation nanodiamond suspensions showing negligible non-linear refraction index in both samples but significative albeit different non-linear absorption behaviour. The HP-HT sample shows absorption decrease increasing the intensity (optical limiting), while the detonation sample shows an inverse behaviour (saturable absorption).

The authors have already done similar work for the detonation nanodiamond samples (refs 49-52, 62) and a previous work for the HP-HT sample (51) showing optical limiting in the nanosecond regime. Here they extend the study to the femtosecond regime and compare the two differently prepared nanodiamond suspensions.

The paper is interesting, but a few points must be clarified

  1. How was the AFM analysis of particle size performed? What software and method (cluster analysis) were employed?
  2. The Raman spectra show differences among the two samples. The Raman peak at 1330 cm-1 is generally present in sp2 carbon nanostructures (nanotubes, graphene) indicating the disorder in the sp2 structure. Here the detonation nanodiamond sample presents larger (in intensity and FWHM) peak. Does this indicate higher amount of defects in the structure? The author should extend the discussion also in view of the conclusion paragraph.
  3. Why HP-HT sample show larger extinction and optical density (equal being the nanoparticle concentration)? Since the perfect diamond structure is transparent (gap higher than 5eV) the linear absorption is expected to increase with the nitrogen concentration, and the author state in the introduction that in the HP-HT it is 100 ppm, while in the detonation samples is 10000ppm. How do they explain the fact lower nitrogen content (100 vs 10000 ppm) lead to larger optical density? Is it due to a scattering effect due to the slightly larger nanoparticles size in the HP-HT sample?
  4. Is this larger extinction spectrum responsible for the absence of saturable absorption in the HP-HT sample?
  5. Following the author lines 322-324 “This means that for both the HP-HT and detonation nanodiamonds, the nonlinear absorption coefficients, i.e., two-photon absorption beta β and saturable absorption coefficients ??? and ?0, linearly depend on the nanoparticles concentration.” For the beta coefficient the authors report the linear dependence (fig.6f, eq. 1) and the fit (line 250) with the constant term (due to water) and the slope (due to the material) which is independent on the nanoparticle concentration and represent the non-linear cross section. Can the author report the same linear dependence for the coefficients ??? and ?0? Do they observe a constant term due to the water or quartz?
  6. The different behaviour among the HP-HT and detonation sample are discussed in terms different band structure and presence of defects. Is there an effect of different nitrogen content (100 vs 10000 ppm) (see point 3)?
  7. In line 333 and following the authors compare the ratio of the saturable to non-saturable losses of detonation samples with that of graphene and CNT concluding that these latter ones are more suitable to be used for passive mode-locking. Can the author comment on the use of detonation and HP-HT nanodiamonds in non-linear optoelectronic devices?

Author Response

Response to Reviewer 1 Comments

Point 1: How was the AFM analysis of particle size performed? What software and method (cluster analysis) were employed?

Response 1: The particle size distribution was attained by processing the atomic-force microscope images with Image Analysis 3.5.0 software based on a combination of the correlation analysis and the method of sections at a certain relative height (see Bykov V. A., Novak V.R., Romanets A.A., Method of selecting local objects on digital surface images. Patent RU 2459251).

Considering the aforementioned, the following text highlighted in yellow has been added to the revised version of the manuscript:

“The particle size distribution was attained by processing the atomic-force microscope images with Image Analysis 3.5.0 software based on the combination of the correlation analysis and the method of sections at a certain relative height.”

Point 2: The Raman spectra show differences among the two samples. The Raman peak at 1330 cm-1 is generally present in sp2 carbon nanostructures (nanotubes, graphene) indicating the disorder in the sp2 structure. Here the detonation nanodiamond sample presents larger (in intensity and FWHM) peak. Does this indicate higher amount of defects in the structure? The author should extend the discussion also in view of the conclusion paragraph.

Response 2: The authors are grateful for this comment. Taking into account the fact that the Raman spectra of nanodiamonds have been specifically studied in a large number of works (see, for example, Refs 34, 53, 54, 69, given in the manuscript, as well as M. Mermoux et al. Diam. Relat. Mater. (2018), doi:10.1016/j.diamond.2018.06.001; C. Pardanaud et al. C-Journal Carbon Res. (2019), doi:10.3390/c5040079), we did not focus on the differences in the Raman spectra of the two types of studied nanodiamonds that are well known to specialists. We agree with the reviewer that the Raman spectra of the HP-HT and detonation diamonds under study also differ significantly. The decomposition of the spectra shown in Fig. 2 (a) shows that the FWHM of diamond lines of HP-HT and detonation nanodiamonds are 8 and 22 cm-1, respectively. Fig. 2 (a) also implies that the Raman line of detonation nanodiamonds is asymmetric with a more pronounced broadening towards lower wavenumbers. All these are in agreement with the above Refs. and, according to Refs. 34 and 69, indicate a greater number of defects present in detonation nanodiamonds.

It should also be noted that the line widths of the X-ray diffraction patterns of the samples under study significantly differ from each other (see Fig. 2 (b)). It is known that the smaller the crystallite size, the larger the corresponding line widths of the X-ray spectra. Processing the measured X-ray diffraction pattern (see Fig. 2 (b)) using TOPAS 4.2 software revealed that the mean crystallite sizes (LVol-IB) of the HP-HT and detonation nanodiamond particles are 8.6 and 2.7 nm, respectively. Taking into account the measured sizes of the studied nanoparticles, this means that the investigated HP-HT nanodiamonds predominantly consist of 1 or 2 single crystallites, while the particles of detonation nanodiamonds are conglomerates, predominantly consisting of 2 to 5 single crystallites. All these findings suggest that the samples of detonation nanodiamonds have a greater number of defects in comparison with the samples of HP-HT nanodiamonds.

To address the reviewer’s remark, the following text highlighted in yellow has been added to the revised version of the manuscript:

The decomposition of the spectra presented in Fig. 2 (a) shows that the full width at half maximum (FWHM) of diamond lines of HP-HT and detonation nanodiamonds are 8 and 22 cm-1, respectively.”

and

“Processing the measured X-ray diffraction pattern using TOPAS 4.2 software revealed that the mean crystallite sizes (LVol-IB) of the HP-HT and detonation nanodiamond particles are 8.6 and 2.7 nm, respectively.

We have also expanded the discussion in the conclusionparagraph as follows:

Indeed, as established above, the average crystallite sizes of HP-HT and detonation nanodiamonds are 8.6 and 2.7 nm, respectively. Taking into account the sizes of the nanoparticles studied (see Fig. 1), this means that HP-HT nanodiamonds are predominantly composed of one or two single crystallites, while detonation nanodiamond particles are conglomerates predominantly composed of two to five single crystallites. The difference in the structures of nanoparticles of the two types of nanodiamonds is also indicated by the difference in their Raman spectra presented in Fig. 2(a). The diamond Raman lines of the HP-HT and detonation nanodiamond spectra with frequency shifts of 1330 and 1329 cm-1, respectively, differ from each other in line width and their shape. The FWHM of the diamond line of detonation nanodiamonds is approximately 2.8 times greater than the corresponding value found for HP-HT nanodiamonds. In addition, the diamond line of detonation nanodiamonds is asymmetric with a more pronounced broadening towards lower wavenumbers. All these findings are in agreement with the literature data, and, according to Refs. 34 and 69, indicate a greater number of defects present in detonation nanodiamonds. Overall, monocrystalline HP-HT nanodiamonds have a uniform structure with the concentration of lattice defects (such as dislocations or twinning) lower than in detonation nanodiamonds containing a diamond core and various lattice defects, transient sp3/sp2 layer, and sp2 surface shell that may carry various surface functional groups [69].

Point 3: Why HP-HT sample show larger extinction and optical density (equal being the nanoparticle concentration)? Since the perfect diamond structure is transparent (gap higher than 5eV) the linear absorption is expected to increase with the nitrogen concentration, and the author state in the introduction that in the HP-HT it is 100 ppm, while in the detonation samples is 10000ppm. How do they explain the fact lower nitrogen content (100 vs 10000 ppm) lead to larger optical density? Is it due to a scattering effect due to the slightly larger nanoparticles size in the HP-HT sample?

Response 3: We are grateful to the reviewer for these questions. In the course of our studies, we also thought about this difference in spectra. The difference in the optical density spectra of detonation and HP-HT nanodiamonds cannot be uniquely attributed to their different nitrogen content, since other mechanisms are leading to light absorption in these materials in a wide spectral range (see references 26, 56, 70, 71, 72). We agree with the reviewer that one of the reasons for this may be their slightly different sizes. However, in our opinion, there is another reason for this that stems from the difference in the "effective" refractive indices of detonation and HP-HT nanodiamonds. HP-HT nanodiamonds are mostly monocrystallites whereas detonation nanodiamond particles are conglomerates consisting of smaller crystallites coated and interconnected by sp2 carbon fractions and sp2/sp3 carbon transition layers. Therefore, we can expect that the refractive index of the scattering particles of HP-HT nanodiamonds is greater than the "effective" refractive index of the scattering particles of detonation nanodiamonds. As is known, the greater the refractive index of scattering particles, the greater their scattering efficiency.

In this regard, we have added the following sentence highlighted in yellow to the text of the revised manuscript:

This can be explained by stronger light scattering in the suspension of HP-HT nanodiamonds caused by the following reasons: (i) the slightly larger average nanoparticle size of the HP-HT nanodiamond suspension compared to the average nanoparticle size of the detonation nanodiamond suspension; (ii) the larger "effective" refractive index of HP-HT nanodiamonds compared to the "effective" refractive index of detonation nanodiamonds due to the difference in their crystal structures.

Point 4: Is this larger extinction spectrum responsible for the absence of saturable absorption in the HP-HT sample?

Response 4: The absence of saturable absorption in the HP-HT sample cannot be explained by the higher value of the extinction coefficient. The absence of saturable absorption in the HP-HT sample may be explained by the features of the energy band structure, which need further separate investigation.

Point 5: Following the author lines 322-324 “This means that for both the HP-HT and detonation nanodiamonds, the nonlinear absorption coefficients, i.e., two-photon absorption beta β and saturable absorption coefficients ??? and ?0, linearly depend on the nanoparticles concentration.” For the beta coefficient the authors report the linear dependence (fig.6f, eq. 1) and the fit (line 250) with the constant term (due to water) and the slope (due to the material) which is independent on the nanoparticle concentration and represent the non-linear cross section. Can the author report the same linear dependence for the coefficients ??? and ?0? Do they observe a constant term due to the water or quartz?

Response 5: As follows from our measurements, there is no saturable absorption of femtosecond laser pulse radiation at 795 nm in water and quartz. Besides, both, quartz and water are transparent at this wavelength. This means that for both water and quartz ??? = 0, ?0 = 0. For the nanosecond laser pulse, we have shown that ???, ?0 linearly depend on the detonation nanodiamond particles concentration (see Ref.49). In addition, the parameters ???, ?0 depend linearly on the particle concentration according to their physical meaning. With the detonation nanoparticle concentration of C=1.25 wt.% we have found ??? = 2.0 cm-1, ?0 = 0.36 cm-1, therefore the following relations ???(cm-1)= 1.6´Ð¡ (wt.%), ?0( cm-1) = 0.24 ´Ð¡ (wt.%) hold true.

Based on the aforementioned, the following sentence of the original manuscript “It should be noted that the obtained values of  and  are defined for the nanoparticle concentration of C = 1.25 wt.% and depend linearly on concentration [49].” has been modified as follows:

“It should be noted that the obtained values of  and  are defined for the nanoparticle concentration of C = 1.25 wt.% and depend linearly on concenration [49], i.e.  ???(cm-1)= 1.6´Ð¡ (wt.%), ?0(cm-1) = 0.24´Ð¡ (wt.%)”

Point 6: The different behaviour among the HP-HT and detonation sample are discussed in terms different band structure and presence of defects. Is there an effect of different nitrogen content (100 vs 10000 ppm) (see point 3)?

Response 6: The difference in the nonlinear optical properties of HP-HT and detonation nanodiamonds originates from the energy absorption spectrum which is defined by the structure and impurities. The nitrogen impurity content can affect their energy spectra, but this is not the only and most likely not the main cause. The answer to this question requires additional investigation, which is beyond the scope of this paper.

Based on this Reviewer’s question the following sentence of the original manuscript “It should be noted that along with the presence of the sp2-carbon [70], the appearance of energy transitions lower than Eg in nanodiamond can be caused by the presence of diamond-like [71] shells, the dimer chains on the surface of diamond nanoparticles [56], as well as the numerous defects and vacancies (see, for example, [29,72–74])” has been rewritten by adding “impurities”:

 “It should be noted that along with the presence of the sp2-carbon [70], the appearance of energy transitions lower than Eg in nanodiamond can be caused by the presence of diamond-like [71] shells, the dimer chains on the surface of diamond nanoparticles [56], as well as the numerous defects, vacancies, and impurities (see, for example, [29,72–74]).

Point 7: In line 333 and following the authors compare the ratio of the saturable to non-saturable losses of detonation samples with that of graphene and CNT concluding that these latter ones are more suitable to be used for passive mode-locking. Can the author comment on the use of detonation and HP-HT nanodiamonds in non-linear optoelectronic devices?

Response 7: Two-photon absorption observed in the aqueous suspension of HP-HT nanodiamonds can be employed for the optical limiting of femtosecond laser pulses, while saturable absorption observed in suspensions of detonation nanodiamonds can be employed for shortening the duration and to remove the pedestal (“tail”) of the laser pulses.

In this regard, we have extended the last two sentences of the article as follows:

The results obtained show that the energy structures of HP-HT and detonation nanodiamonds differ from each other significantly. Aqueous suspensions of HP-HT nanodiamonds can be used for the optical limiting of high-power femtosecond laser pulses while suspensions of detonation nanodiamonds possess properties of the saturable absorber and can be employed for shortening the duration and to remove the pedestal (“tail”) of the laser pulses.

Reviewer 2 Report

Authors discuss the nonlinear optical properties of two kinds of particle nanodiamonds diluted in water: HP-HT and detonation nanodiamonds. They do not observe significant nonlinear refraction and different type of nonlinear absorption: two-photon absorption (HP-HT nanodiamonds) and nonlinear saturable absorption (detonation nanodiamonds). In both cases, hey fit the experimental observed z-scan aperture transmission, to obtain the different parameters that define the absorption coefficients. This work is a continuation of the work presented in ref. 49.

The manuscript is well written, author presents sufficient evidence of the nonlinear behavior and discussion seams correct. I recommend publishing this work in Applied Sciences.

Some suggestions:

i. Have you studied whether particle size influences the nonlinear properties?

ii. It is strange that with nonlinear absorption, the nonlinear refraction is negligible. Could you comment on this a little?

iii. For HP-HT nanodiamonds, the absorption increases with intensity while for detonation nanodiamonds the absorption decreases with intensity. Therefore, it would be interesting to test how the two phenomena compete if we mix the two types of diamonds.

Author Response

Response to Reviewer 2 Comments

Point 1: Have you studied whether particle size influences the nonlinear properties?

Response 1: We have studied the effect of the detonation nanodiamond particle size on the nonlinear optical properties. Specifically, in the paper by Vanyukov V. V., Mogileva T. N., Mikheev G. M., Puzyr A. P., Bondar V. S., Svirko Y. P. Size effect on the optical limiting in suspensions of detonation nanodiamond clusters / Applied Optics, vol. 52, issue 18, pp. 4123-4130 we have presented the findings of the study of the effect of nanoparticle sizes on the nonlinear scattering, which results in the optical limiting of nanosecond laser pulses.

The influence of nanoparticle sizes on two-photon absorption in HP-HT nanodiamond suspensions and on the saturable absorption in detonation nanodiamond suspensions in the femtosecond domain is beyond the scope of this work and is the subject of further studies.

Point 2: It is strange that with nonlinear absorption, the nonlinear refraction is negligible. Could you comment on this a little?

Response 2: In our experiments, the nonlinear refraction in the nanodiamond suspension due to heat absorption does not manifest itself. This is explained by the fact that the nonlinear absorption observed in the experiments is weak. For example, in the HP-HT nanodiamond suspension at a nanoparticle concentration of C=1.65 wt.% at a laser pulse energy of 200 nJ, the nonlinear absorption is only about 3% (see Fig. 6(a)), and in the suspension of detonation nanodiamonds at the concentration of nanoparticles C=1.25 wt.% at the energy of laser pulses 360 nJ the nonlinear bleaching does not exceed 4%. 

In this regard, we have added the following sentence to the text of the revised manuscript:

It should be added that in our experiments, despite the nonlinear absorption, the nonlinear refraction in the nanodiamond suspension due to heat absorption does not manifest itself. This is explained by the fact that the nonlinear absorption observed in the experiments is weak. For example, in the HP-HT nanodiamond suspension at a nanoparticle concentration of C=1.65 wt.% at a laser pulse energy of 200 nJ, the nonlinear absorption is only about 3% (see Fig. 6(a)), and in the suspension of detonation nanodiamonds at the concentration of nanoparticles C=1.25 wt.% at the energy of laser pulses 360 nJ the nonlinear absorption does not exceed 4%.

Point 3: For HP-HT nanodiamonds, the absorption increases with intensity while for detonation nanodiamonds the absorption decreases with intensity. Therefore, it would be interesting to test how the two phenomena compete if we mix the two types of diamonds.

Response 3: Indeed, the proposed idea of the experiment is interesting. However, the goal of our study was to experimentally demonstrate the difference in the nonlinear optical properties of HP-HT and detonation nanodiamonds under femtosecond excitation. For this reason, we were strictly avoiding mixing suspensions of HP-HT and detonation nanodiamonds with each other in our experiments.

Round 2

Reviewer 1 Report

In their revised version the authors have addressed all the points raised in my report. The paper deserves to be accepted in the present form.